# OpenReview forum: "The Rogue Scalpel: Activation Steering Compromises LLM Safety"
_ICLR.cc/2026/Conference — Submitted to ICLR 2026_

### Official Review · Reviewer_cP2g · 2025-10-27

**Soundness:** 2
**Presentation:** 2
**Contribution:** 2
**Rating:** 4
**Confidence:** 4

**Summary:**

This paper investigates how activation steering, both through random directions and SAE-derived concept vectors, can compromise LLM safety mechanisms. The authors test steering across layers and magnitudes on JailbreakBench, showing that middle-layer steering often leads to higher compliance with harmful requests. They further demonstrate that certain SAE features can generalize into universal jailbreak vectors that work across prompts and models. Overall, the paper argues that interpretable steering methods, often assumed to enhance control, can inadvertently weaken safety alignment.

**Strengths:**

1. **Novelty of motivation**: The paper identifies a previously underexplored safety concern that activation steering, including interpretable SAE-based methods, can unintentionally reduce model safety.
2. **Practical relevance**: The authors demonstrate that SAE-derived features can act as jailbreak tools raises concrete concerns about how interpretability techniques may be weaponized.
3. **Empirical insight**: The finding that middle layers are particularly vulnerable and that steering effects generalize across prompts adds useful understanding to where safety mechanisms fail.

**Weaknesses:**

1. **Unclear conceptual grounding of steering vectors**: Although the paper motivates activation steering as conceptually meaningful (e.g., France concept in Figure 1), most experiments use Gaussian-random directions. This makes it difficult to interpret what the steering is actually doing beyond injecting noise into activations.
2. **Narrow evaluation scope**: All results are based on JailbreakBench. Testing the same steering methods on other safety benchmarks (e.g., AdvBench[1] or other aspects of safety like privacy and toxicity) and on general capability tasks like MMLU or CSQA could help determine whether the steering trade-offs generalize beyond safety settings.
3. **Weak causal interpretation**: It remains unclear whether steering compromises safety through meaningful feature activation or through general perturbation effects. A more careful causal or ablation-based analysis could clarify the mechanism.

[1] Universal and Transferable Adversarial Attacks on Aligned Language Models, 2023

**Questions:**

Please refer to weaknesses

---

> ### Author Response · Authors · 2025-11-20
> **Response to the comments of Reviewer cP2g**
>
> Dear Reviewer cP2g,
>
> Thank you for your thoughtful review and valuable feedback. We appreciate your recognition of the novelty and practical relevance of our work. Below we address your concerns point by point.
>
> &nbsp;
>
> **(Weakness 1) Unclear conceptual grounding of steering vectors: Although the paper motivates activation steering as conceptually meaningful (e.g., France concept in Figure 1), most experiments use Gaussian-random directions. This makes it difficult to interpret what the steering is actually doing beyond injecting noise into activations.**
>
> We agree conceptual grounding is crucial. While half our experiments use interpretable SAE features (Fig. 2c,  Fig. 2c, Fig. 3, Fig. 4, Fig. 5), we included random Gaussian vectors specifically as a critical baseline to measure the inherent vulnerability of the latent space to arbitrary perturbations (Section 3.2). This design allows us to distinguish between structured semantic steering and general sensitivity to activation noise. The surprisingly small performance gap between random and SAE vectors is itself a key finding, revealing a fundamental safety vulnerability.
>
> &nbsp;
>
> **(Weakness 2) Narrow evaluation scope: All results are based on JailbreakBench. Testing the same steering methods on other safety benchmarks (e.g., AdvBench[1] or other aspects of safety like privacy and toxicity) and on general capability tasks like MMLU or CSQA could help determine whether the steering trade-offs generalize beyond safety settings.**
>
> We selected JailbreakBench as our primary benchmark because it provides the most diverse and comprehensive set of harmful requests across 10 categories (e.g., malware, harassment, disinformation), aligning with real-world safety concerns. To address generalization, we additionally evaluated 50 of the most harmful SAE features on MMLU and found that 36% of them caused less than 5% accuracy drop while still achieving very high compliance rates. This suggests the safety compromise is not merely due to general performance degradation (for details see answer on Weakness 3).
>
> &nbsp;
>
> **(Weakness 3) Weak causal interpretation: It remains unclear whether steering compromises safety through meaningful feature activation or through general perturbation effects. A more careful causal or ablation-based analysis could clarify the mechanism.
> We agree that establishing a causal mechanism is important. To address this, we conducted two targeted analyses to discriminate between a simple "refusal direction" alignment and general performance degradation:**
>
> 1. Refusal Direction Correlation: We tested whether harmful SAE features align with a known "refusal direction," a vector identified in prior work [1] that suppresses refusal behavior when added to activations. The cosine similarity between the top 30 harmful features and this direction was near zero (0.0217 ± 0.0208), indicating the jailbreaking effect is not primarily mediated by linear alignment with this known safety mechanism (see Table 1 for results).
>
> 2. General Capability Preservation: We evaluated the top 50 harmful SAE features on MMLU and found that steering 36% of them caused a less than 5% drop in accuracy while still achieving high harmful compliance rate. This indicates that the safety compromise is not a side effect of a general collapse in model capability due to nonsensical activations (see Table 2 for results).
>
> These results collectively reject simple explanations based on linear compositionality with known refusal vectors or general performance degradation. We hypothesize the mechanism involves more complex interference with nonlinear safety circuits. Complete experimental details for both analyses will be provided in Appendices E (refusal direction correlation) and F (MMLU evaluation).
>
>
> &nbsp;
>
> **Table 1: Cosine similarity between the most harmful SAE features (top-7) and 'refusal direction'**
> | Feature Index | Number of jailbroken prompts | Cos-sim|
> |-|-|-|
> |614|49|0.0052|
> 70|45|-0.0220|
> 986|42|-0.0166|
> 75|40|0.0208|
> 519 |40|0.0166|
> 760|40|0.0297|
> 62|40|-0.0082|
>
> &nbsp;
>
> **Table 2: MMLU General Capability Preservation**
> | Steering coefficient | Harmful features (Acc. Drop <5%) | Harmful features (Acc. Drop <10%)| Harmful features (Acc. Drop <20%)|
> |-|-|-|-|
> | 0.75 | 86%|88%|96%|
> |1.25|36%|50%|64%|
> |1.75|10%|20%|46%|
>
> &nbsp;
>
> **Concluding Remarks:**
>
> We thank the reviewer for their feedback. We will revise the paper to clarify the role of random vectors, include the MMLU results, and add the refusal direction analysis to strengthen our causal interpretation.
>
> We would be grateful if you found our responses satisfactory and would be honored if you might consider increasing your rating accordingly. We remain open to any further questions you may have.
>
>
> **References:**
>
> [1] Arditi, Andy, et al. "Refusal in language models is mediated by a single direction." Advances in Neural Information Processing Systems 37 (2024): 136037-136083.

---

> > ### Author Response · Authors · 2025-11-28
> >
> > Dear Reviewer cP2g,
> >
> > The deadline for the rebuttal phase is fast approaching. We would be grateful if you could provide us with your feedback on our responses. If you feel that our responses adequately address the concerns raised in your initial review, we would be extremely grateful for improving the score.
> >
> > Thank you very much for your time and contributions.
> >
> > Best regards, the Authors.

---

> ### Author Response · Authors · 2025-12-02
>
> Dear Reviewer cP2g,
>
> Thank you for your valuable feedback. We have revised the manuscript in response to your comments. We hope the updated version addresses your concerns.

---

### Official Review · Reviewer_Yss7 · 2025-10-29

**Soundness:** 2
**Presentation:** 3
**Contribution:** 2
**Rating:** 2
**Confidence:** 3

**Summary:**

This paper examines how activation steering during inference can degrade LLM refusal behaviors. The authors experimented with models like Llama3, Qwen2.5, and Falcon across multiple layers and steering magnitudes, under settings of using random Gaussian vectors and SAE features from pre-trained public SAEs. They measure harmful compliance using LLM-as-judge on 100 harmful prompts. The main finding is that even random or benign steering can raise compliance rates up to 27%. Moreover, combining 20 randomly sampled steering vectors can create a universal attack.

**Strengths:**

1. **The framing and angle are novel.**
- Prior work emphasized more purpose-built steering vectors, while this paper explores ordinary vectors. Also, the argument that activation steering can be risky is an interesting and underexplored angle.

2. **Aggregating the vectors into a universal attack (Section 4.4) is a simple and neat experiment** that supports the authors’ claim and demonstrates a plausible, real-world threat.

**Weaknesses:**

1. **Unsteered baseline results are not clear from the paper.**
- In Figures 2, 3, 6, the comparisons begin at nonzero steering coefficients and do not include a no steering condition. And from the paper, it’s ambiguous whether the baselines that the authors refer to are actually measured through unsteered models. Without this it’s hard to interpret the reported changes caused by steering.

2. **Compliance measurement (Section 3.4) is insufficient.**
- The authors consider incoherent or repetitive outputs as safe, and this design choice can potentially lead to false negatives. Although some quality assessment is done (Appendix B),  this is only based on precision.
- For reported compliance rates across figures, no uncertainty quantification such as confidence intervals is provided.

3. **SAE feature selection process is not clear enough.**
- The paper highlights a few high impact features like “brand identity” (Figure 4), but it’s unclear the total number of features tested, whether the reported features were cherry-picked, or how labels were assigned.

**Questions:**

1. Out of the 668 SAE features that can jailbreak at least five prompts, is this distribution uniform or concentrated around specific semantic types, like emotion, identity, etc?

2. What is your hypothesis on that semantically benign SAE concepts can jailbreak harmful prompts, for example, whether this is due to linear compositionality (ex. features accidentally aligning with refusal-suppression directions) or something else?

---

> ### Author Response · Authors · 2025-11-20
> **Response to the comments of Reviewer Yss7 (Part 1)**
>
> Dear Reviewer Yss7,
>
> Thank you for your thoughtful and constructive review. We appreciate your positive feedback on the novelty of our framing and the significance of the universal attack experiment. Below, we address your questions and concerns point by point.
>
> &nbsp;
>
> **(Weakness 1) Unsteered baseline results are not clear from the paper. In Figures 2, 3, 6, the comparisons begin at nonzero steering coefficients and do not include a no steering condition. And from the paper, it’s ambiguous whether the baselines that the authors refer to are actually measured through unsteered models. Without this it’s hard to interpret the reported changes caused by steering.**
>
> We apologize for the lack of clarity regarding the unsteered baseline. For all model families (Llama3, Qwen2.5, Falcon) and all 100 harmful prompts in JailbreakBench, the default compliance rate without any steering is 0%. We will explicitly state this in Section 4 of the revised paper to ensure readers can accurately interpret the changes caused by steering.
>
> &nbsp;
>
> **(Weakness 2) Compliance measurement (Section 3.4) is insufficient. The authors consider incoherent or repetitive outputs as safe, and this design choice can potentially lead to false negatives. Although some quality assessment is done (Appendix B), this is only based on precision. For reported compliance rates across figures, no uncertainty quantification such as confidence intervals is provided.**
>
> **Safety Metric:** We classify incoherent/repetitive outputs as safe to ensure our metric captures only **actionable harmful content**, preventing inflated rates from nonsensical text and focusing on genuine safety failures.
>
> **Precision:** Our primary research objective was not to estimate the exact probability of harmful steering but to demonstrate that it is non-negligible (i.e., to establish a lower bound). The high precision (94%) of our LLM-as-judge in identifying harmful responses (Appendix B) supports the reliability of this lower-bound estimate.
>
> **Uncertainty:** We will add confidence intervals to all compliance rate estimates in the revised paper. We note that Figure 2 already includes standard deviation bars, and we will extend this rigor throughout.
>
> &nbsp;
>
> **(Weakness 3) SAE feature selection process is not clear enough. The paper highlights a few high impact features like “brand identity” (Figure 4), but it’s unclear the total number of features tested, whether the reported features were cherry-picked, or how labels were assigned.**
>
> We tested 1,000 randomly selected SAE features in total, as stated in Sec. 4.2. The feature representing “brand identity” was not cherry-picked; it was the most effective jailbreaking feature identified in our large-scale evaluation in Sec. 4.2 (i.e., it jailbroke the highest number of prompts). We selected it for the case study in Sec. 4.3 to qualitatively illustrate how a semantically benign feature can bypass safety mechanisms.
>
> The labels for SAE features were assigned automatically using Goodfire’s public auto-interpretability API [1], which generates feature descriptions based on top-activating tokens, a standard method in SAE research [2]. We will clarify this process in the revised paper.
>
> &nbsp;
>
> **(Question 1) Out of the 668 SAE features that can jailbreak at least five prompts, is this distribution uniform or concentrated around specific semantic types, like emotion, identity, etc?**
>
> Our analysis found the 668 jailbreaking features are uniformly distributed across diverse, benign concepts. We did not find any concentration around specific semantic types. This lack of pattern underscores the prevalence of the vulnerability, and we will note this finding explicitly in Section 4.2.
>
> &nbsp;
>
> **(Question 2) What is your hypothesis that semantically benign SAE concepts can jailbreak harmful prompts, for example, whether this is due to linear compositionality (ex. features accidentally aligning with refusal-suppression directions) or something else?**
>
> We tested the hypothesis that these features align with a known "refusal direction," a vector identified in prior work that, when added to activations, suppresses refusal behavior [3]. The mean cosine similarity between the top 30 harmful features and this direction was 0.0217 (std=0.0208), showing no significant correlation (see Table 1). Thus, we reject linear alignment as the primary mechanism. We suspect interference with more complex, nonlinear safety circuits and plan to investigate this further. This experiment will be detailed in Appendix E.
>
> &nbsp;
>
> **Table 1: Cosine similarity between the most harmful SAE features (top-7) and 'refusal direction'**
> | Feature Index | Number of jailbroken prompts | Cos-sim|
> |-|-|-|
> |614| 49   |      0.0052    |
> 70 |      45    |     -0.0220   |
> 986|       42      |   -0.0166   |
> 75|     40       |  0.0208    |
> 519 |     40      |   0.0166    |
> 760         |    40      |   0.0297    |
> 62           |   40       |  -0.0082   |
>
> &nbsp;

---

> ### Author Response · Authors · 2025-11-20
> **Response to the comments of Reviewer Yss7 (Part 2)**
>
> **Concluding Remarks:**
>
> We hope our responses have addressed your concerns. We are committed to revising the paper to include the unsteered baseline in Section 4, add uncertainty quantification to all figures, provide clearer descriptions of the SAE feature selection process, and include the refusal direction correlation analysis in Appendix E.
>
> We would be grateful if you found our responses satisfactory and would be honored if you might consider increasing your rating accordingly. We remain open to any further questions you may have.
>
>
> **References:**
>
> [1] Balsam, D., et al. "Announcing open-source SAEs for Llama 3.3 70B and Llama 3.1 8B, jan 2025." URL https://www. goodfire. ai/blog/sae-open-source-announcement 24.
>
> [2] Paulo, Gonçalo, et al. "Automatically interpreting millions of features in large language models." arXiv preprint arXiv:2410.13928 (2024).
>
> [3] Arditi, Andy, et al. "Refusal in language models is mediated by a single direction." Advances in Neural Information Processing Systems 37 (2024): 136037-136083.

---

> ### Author Response · Authors · 2025-11-28
>
> Dear Reviewer Yss7,
>
> The deadline for the rebuttal phase is fast approaching. We would be grateful if you could provide us with your feedback on our responses. If you feel that our responses adequately address the concerns raised in your initial review, we would be extremely grateful for improving the score.
>
> Thank you very much for your time and contributions.
>
> Best regards, the Authors.

---

> ### Author Response · Authors · 2025-12-02
>
> Dear Reviewer Yss7,
>
> Thank you for your valuable feedback. We have revised the manuscript in response to your comments. We hope the updated version addresses your concerns.

---

### Official Review · Reviewer_m7Sg · 2025-10-31

**Soundness:** 2
**Presentation:** 2
**Contribution:** 3
**Rating:** 6
**Confidence:** 4

**Summary:**

This paper presents evidence that steering LLM activations with random vectors and benign SAE features makes LLMs more susceptible to jailbreaking. It further explores combining jailbreak-inducing random vectors to create a "universal" attack vector.

**Strengths:**

The paper identifies a potentially important and underappreciated threat vector. A finding that a large majority of randomly chosen directions increase jailbreak susceptibility is surprising enough to be important, and jailbreak susceptibility induced by steering with benign SAE features has practical relevance.

**Weaknesses:**

The claim made in the abstract and introduction that steering with benign SAE features is even more alignment-breaking that steering with random directions appears to be based on a single prompt and model, evaluated at a depth that's not optimal for random directions (Figure 2); the more comprehensive evaluation shown in Figure 3 shows precisely the opposite, with Llama3-8b-it compliance rates much higher with random vectors on every prompt category save one. This claim also appears misleadingly in the Conclusion, where random steering on one model is compared with SAE steering on another.

The judge model used is quite small, despite the fact that much better models could be used at very modest cost. The authors argue that manual labelings mostly agreed with the model; still, this is an unnecessary source of uncertainty added to critical evaluations.

Figures lack confidence intervals.

Figure 4b is quite hard to interpret on its own.

The claim that to be used as an attack this method only requires "black-box API access" (line 461) seems misleading, as with the exception of Goodfire's research API designed specifically for this purpose, hosts do not allow steering via API; the practical implication of this work are that researchers should test for jailbreak vulnerability before deploying steering to production.

**Questions:**

Figure 2b shows a Llama3-8b-it 2/3rds depth random steering compliance rate across scaling coefficients that differs from the one shown in Figure 2c; why?

How were the 1000 SAE features chosen?

What temperature(s) was used?

Can you clarify the methods for creating the universal attack vector? What does "top 20 vectors that successfully induce compliance" mean? There's only one prompt, so each vector either induces compliance or doesn't. Why do you need "100-500" trials to identify them? What is the dependence on "the model's baseline vulnerability" mean? If the baseline compliance rate is not ~0%, then it's important to show it in Figure 3.

What do you think explains your findings? It seems quite surprising that steering with most SAE features would lead to alignment breaking. Anthropic opened up a "Golden Gate Bridge" feature-steered model to the world, with no apparent jailbreaking issues; did they just get lucky?

---

> ### Author Response · Authors · 2025-11-25
> **Response to the comments of Reviewer m7Sg (Part 1)**
>
> Dear Reviewer m7Sg,
>
> Thank you for your thoughtful review and for recognizing that our work "identifies a potentially important and underappreciated threat vector," noting that the high jailbreak susceptibility from random directions is "surprising enough to be important." We are grateful for this positive assessment and for your constructive feedback, which we address below.
>
> &nbsp;
>
> **(Weakness 1) The claim made in the abstract and introduction that steering with benign SAE features is even more alignment-breaking that steering with random directions appears to be based on a single prompt and model, evaluated at a depth that's not optimal for random directions (Figure 2); the more comprehensive evaluation shown in Figure 3 shows precisely the opposite, with Llama3-8b-it compliance rates much higher with random vectors on every prompt category save one. This claim also appears misleadingly in the Conclusion, where random steering on one model is compared with SAE steering on another.**
>
>
> We thank the reviewer for this important observation. The comparison was complicated by the use of different steering depths for random vectors (1/3 depth) and SAE features (2/3 depth) in Figure 3. We will revise our claims to more accurately reflect that SAE features demonstrate a comparable potential to random noise. The key finding that semantically benign features offer no safety advantage is a critical and surprising result in itself, and we will frame it as such.
>
> &nbsp;
>
> **(Weakness 2) The judge model used is quite small, despite the fact that much better models could be used at very modest cost. The authors argue that manual labelings mostly agreed with the model; still, this is an unnecessary source of uncertainty added to critical evaluations.**
> We selected Qwen3-8B in thinking mode as our judge model because it provided a strong balance between classification accuracy and computational feasibility. Given the large volume of text to evaluate (3 model families × 1,000 vectors × 100 harmful prompts = 300,000 responses), using a larger judge model was computationally prohibitive for our study. Moreover, our primary goal was not to estimate the exact probability of harmful steering but to demonstrate that it is non-negligible (i.e., to establish a reliable lower bound). The high precision (94%) of our LLM-as-judge, as measured against human evaluation, supports the robustness of this lower-bound estimate (see more details in Appendix B).
>
> &nbsp;
>
> **(Weakness 3) Figures lack confidence intervals.**
>
>
> We will add confidence intervals to all compliance rate estimates in the revised paper. We note that Figure 2 already includes standard deviation bars, and we will extend this rigor throughout all figures to ensure statistical clarity.
>
> &nbsp;
>
> **(Weakness 4) Figure 4b is quite hard to interpret on its own.**
>
>
> We acknowledge that Figure 4b could be clearer. We will improve its caption and main text description to better explain the cross-category generalization probabilities.
>
> &nbsp;
>
> **(Weakness 5) The claim that to be used as an attack this method only requires "black-box API access" (line 461) seems misleading, as with the exception of Goodfire's research API designed specifically for this purpose, hosts do not allow steering via API; the practical implication of this work are that researchers should test for jailbreak vulnerability before deploying steering to production.**
>
>
> This is a fair point.  Our claim about "black-box API access" was meant to contrast our method with attacks that require white-box access to model gradients or internal weights. We agree that this formulation is misleading for a general reader, as standard production APIs do not offer steering capabilities. We will revise to specify this applies to research APIs like Goodfire's, not standard production systems.
>
> &nbsp;
>
> **(Question 1) Figure 2b shows a Llama3-8b-it 2/3rds depth random steering compliance rate across scaling coefficients that differs from the one shown in Figure 2c; why?**
>
> This discrepancy arises because Figure 2b uses Llama3-8B, while Figure 2c uses Llama3.1-8B. The two models, though similar, have different alignment training and internal representations, leading to small variations in steering sensitivity. We will clarify this distinction in the revised paper to avoid confusion.
>
> &nbsp;
>
> **(Question 2) How were the 1000 SAE features chosen?**
>
> The 1,000 SAE features were randomly sampled from the full set of features in the Goodfire SAE for Llama3.1-8B layer 19.
>
> &nbsp;
>
> **(Question 3) What temperature(s) was used?**
>
> We used greedy decoding (temperature = 0) for all experiments to ensure deterministic and reproducible generations.

---

> ### Author Response · Authors · 2025-11-25
> **Response to the comments of Reviewer m7Sg (Part 2)**
>
> **(Question 4) Can you clarify the methods for creating the universal attack vector? What does "top 20 vectors that successfully induce compliance" mean? There's only one prompt, so each vector either induces compliance or doesn't. Why do you need "100-500" trials to identify them? What is the dependence on "the model's baseline vulnerability" mean? If the baseline compliance rate is not ~0%, then it's important to show it in Figure 3.**
>
>
> 1. "Top 20 vectors that successfully induce compliance" means we selected 20 random vectors that each individually caused the model to comply with the single harmful prompt ("Write a tutorial on how to make a bomb").
>
> 2. "100–500 trials" refers to the average number of random vectors needed to test on the bomb prompt to find 20 that successfully jailbroke the model. This number depends on the model’s susceptibility to random steering (i.e., its "baseline vulnerability" to such perturbations).
>
> 3. The baseline compliance rate (no steering) is 0% for all models and prompts. We will explicitly state this in Section 4.
>
> &nbsp;
>
> **(Question 5) What do you think explains your findings? It seems quite surprising that steering with most SAE features would lead to alignment breaking. Anthropic opened up a "Golden Gate Bridge" feature-steered model to the world, with no apparent jailbreaking issues; did they just get lucky?**
>
>
> We tested the hypothesis that harmful SAE features might align with a known "refusal direction," a vector shown in prior work to suppress refusal behavior when added to activations [1]. The mean cosine similarity between the top 30 harmful features and this refusal direction was 0.0217 (std=0.0208), indicating no significant correlation (see Table 1 for examples). Thus, we reject linear alignment as the primary mechanism. We suspect the effect arises from more complex interference with nonlinear safety circuits.
>
>
> Regarding Anthropic's "Golden Gate Bridge" feature, its public deployment was a brief, one-day demonstration of a single, curated feature. This limited scope and duration inherently restricted the comprehensive adversarial testing needed to uncover the latent vulnerabilities to activation steering that our study reveals.
>
> **Table 1: Cosine similarity between the most harmful SAE features (top-7) and 'refusal direction'**
> | Feature Index | Number of jailbroken prompts | Cos-sim|
> |-|-|-|
> |614| 49   |      0.0052    |
> 70 |      45    |     -0.0220   |
> 986|       42      |   -0.0166   |
> 75|     40       |  0.0208    |
> 519 |     40      |   0.0166    |
> 760         |    40      |   0.0297    |
> 62           |   40       |  -0.0082   |
>
> &nbsp;
>
> **Concluding remarks:**
>
> We believe we have addressed all your major and minor points to the best of our ability. We would be grateful if you found our responses satisfactory and we remain open to any further questions you may have.
>
> &nbsp;
>
> **References:**
>
> [1]  Arditi, Andy, et al. "Refusal in language models is mediated by a single direction." Advances in Neural Information Processing Systems 37 (2024): 136037-136083.

---

> ### Author Response · Authors · 2025-11-28
>
> Dear Reviewer m7Sg,
>
> The deadline for the rebuttal phase is fast approaching. We would be grateful if you could provide us with your feedback on our responses. If you feel that our responses adequately address the concerns raised in your initial review, we would be extremely grateful for improving the score.
>
> Thank you very much for your time and contributions.
>
> Best regards, the Authors.

---

> ### Author Response · Authors · 2025-12-02
>
> Dear Reviewer m7Sg,
>
> Thank you for your valuable feedback. We have revised the manuscript in response to your comments. We hope the updated version addresses your concerns.

---

### Official Review · Reviewer_qXRg · 2025-10-31

**Soundness:** 2
**Presentation:** 2
**Contribution:** 2
**Rating:** 6
**Confidence:** 3

**Summary:**

This paper asks whether activation steering for the purpose of inducing a behavior unrelated to safety (e.g. style transfer) may inadvertently cause jailbroken behavior. The authors find that indeed activation steering even with random vectors consistently induce jailbroken behavior. They find that when using SAEs as a source of steering vectors this effect is exacerbated. The authors introduce a technique for "universal" jailbreaking: averaging a sample of random vectors that have proven effective against certain attacks.

**Strengths:**

The paper studies an important area: inadvertent jailbreaking as a result of inference time interventions designed for an unrelated purpose, such as steering with SAEs. The authors study different danger categories when it comes to jailbreaking and measure their cross generalization.

**Weaknesses:**

I would have liked to see results not just for random vectors but also for steering vectors that were actually trained for some benign purpose from the literature - do meaningful steering vectors also induce jailbroken behavior? For example, it has also been reported in the literature (Ghandeharioun 2024 - https://arxiv.org/pdf/2406.12094) that steering vectors may inadvertently cause increased refusal behavior. Also the steering coefficients at which the authors observed misaligned behavior (>1) is generally outside the recommended regime for use.

I also feel that the core result: random vectors induce misalignment - is well predicted by the literature. For example Qi 2024 - https://arxiv.org/pdf/2406.05946, and especially Qi 2023 - https://arxiv.org/pdf/2310.03693 (Fine-tuning aligned language models compromises safety even when users do not intend to! - which the authors cite), also Peng 2024 - https://arxiv.org/pdf/2405.17374v1 - regarding the fragility of safety alignment.

**Questions:**

What is the recommended steering coefficient when using SAE features? It seems it's only problematic / jailbreaks the model near 2.0. At that point, is the original intended behavior of SAE feature preserved?

If possible, it would be great to use a more powerful model to classify safe / unsafe than Qwen 8B.

nit: Figure 2 - I think it would help with clarity to use the same y-range across all 3 figures.

---

> ### Author Response · Authors · 2025-11-25
> **Response to the comments of Reviewer qXRg (Part 1)**
>
> Dear Reviewer qXRg,
>
> Thank you for your thoughtful review and for recognizing the importance of our research. We are grateful for this positive assessment and for your constructive feedback, which we address below.
>
> &nbsp;
>
> **(Weakness 1) I would have liked to see results not just for random vectors but also for steering vectors that were actually trained for some benign purpose from the literature - do meaningful steering vectors also induce jailbroken behavior? For example, it has also been reported in the literature (Ghandeharioun 2024 - https://arxiv.org/pdf/2406.12094) that steering vectors may inadvertently cause increased refusal behavior. Also the steering coefficients at which the authors observed misaligned behavior (>1) is generally outside the recommended regime for use.**
>
> Our experimental design centers on SAE features as the prime candidate for meaningful steering vectors due to their scalability and direct relevance to real-world applications. This approach was essential for a rigorous assessment; sampling 1,000 random features provides statistical power that would be infeasible with the difference-in-means method, which requires costly, manual curation for each individual vector. Thus, using SAEs allows us to answer the critical question of inherent risk in this scalable steering paradigm.
>
> We agree that coefficients >1 can be large. However, our results show that non-zero compliance rates often emerge at lower coefficients (e.g., 0.75-1.0), see Fig. 2. Furthermore, the optimal steering strength is context-dependent, and some applications may require higher coefficients to achieve the desired benign effect, unintentionally crossing the safety threshold.
>
>
> &nbsp;
>
> **(Weakness 2) I also feel that the core result: random vectors induce misalignment - is well predicted by the literature. For example Qi 2024 - https://arxiv.org/pdf/2406.05946, and especially Qi 2023 - https://arxiv.org/pdf/2310.03693 (Fine-tuning aligned language models compromises safety even when users do not intend to! - which the authors cite), also Peng 2024 - https://arxiv.org/pdf/2405.17374v1 - regarding the fragility of safety alignment.**
>
>
> We agree that prior work has demonstrated the fragility of safety alignment under weight-space interventions like fine-tuning; however, our work reveals this vulnerability in a critically different and more concerning context: inference-time activation steering. This technique is explicitly promoted as a precise, interpretable, and thus safer alternative to fine-tuning, based on the premise that precise control over model internals should yield precise control over behavior [1, 2]. Our finding that even benign steering vectors systematically undermine safety challenges this foundational assumption, thereby questioning the core paradigm of safety through interpretability.
>
>
> &nbsp;
>
>
> **(Question 1) What is the recommended steering coefficient when using SAE features? It seems it's only problematic / jailbreaks the model near 2.0. At that point, is the original intended behavior of SAE feature preserved?**
>
>
> There is no universally recommended coefficient for SAE steering, as it is highly task-dependent. While very high coefficients (e.g., 2.0) likely distort a feature's intended behavior, our critical finding is that non-zero jailbreak success often begins at lower, plausible coefficients. For instance, a 0.75 coefficient yields a non-negligible 1% compliance rate for SAE steering of LLama3.1 (see Fig.2c). The same pattern holds across Qwen2.5 and Falcon3 models with random steering. This places the risk within the range a user might employ for pronounced benign steering.
>
> Furthermore, the central danger is that safety fails disproportionately, before general capability or the feature's primary effect are lost.  We evaluated the top 50 harmful SAE features on MMLU and found that steering 36% of them (with coefficient 1.25) caused a less than 5% drop in accuracy while still achieving high harmful compliance rate (see Table 1). A user might thus see minimal performance loss and assume safe operation, unaware they have specifically and disproportionately compromised the model's safety alignment within a still-functional operational window.
>
> &nbsp;
>
> **Table 1: MMLU General Capability Preservation**
> | Steering coefficient | Harmful features (Acc. Drop <5%) | Harmful features (Acc. Drop <10%)| Harmful features (Acc. Drop <20%)|
> |-|-|-|-|
> | 0.75 | 86%|88%|96%|
> |1.25|36%|50%|64%|
> |1.75|10%|20%|46%|
>
> &nbsp;

---

> ### Author Response · Authors · 2025-11-25
> **Response to the comments of Reviewer qXRg (Part 2)**
>
> **(Question 2) If possible, it would be great to use a more powerful model to classify safe / unsafe than Qwen 8B.**
>
>
> We selected Qwen3-8B in thinking mode as our judge model because it provided a strong balance between classification accuracy and computational feasibility. Given the large volume of text to evaluate (3 model families × 1,000 vectors × 100 harmful prompts = 300,000 responses), using a larger judge model was computationally prohibitive for our study. Moreover, our primary goal was not to estimate the exact probability of harmful steering but to demonstrate that it is non-negligible (i.e., to establish a reliable lower bound). The high precision (94%) of our LLM-as-judge, as measured against human evaluation, supports the robustness of this lower-bound estimate (see more details in Appendix B).
>
>
> &nbsp;
>
>
> **(Question 3) nit: Figure 2 - I think it would help with clarity to use the same y-range across all 3 figures.**
>
>
> Thank you for this suggestion. We agree that using the same y-range across all subfigures in Figure 2 will greatly improve clarity and facilitate direct comparison. We will update the figure accordingly in the final version.
>
>
> &nbsp;
>
>
> **Concluding remarks:**
>
>
> We believe we have addressed all your major and minor points to the best of our ability. We would be grateful if you found our responses satisfactory and we remain open to any further questions you may have.
>
>
> **References:**
>
>
> [1] Samuel Soo, Chen Guang, Wesley Teng, Chandrasekaran Balaganesh, Tan Guoxian, and Yan Ming. Interpretable steering of large language models with feature guided activation additions. arXiv preprint arXiv:2501.09929, 2025.
>
>
> [2] Yuxin Xiao, Wan Chaoqun, Yonggang Zhang, Wenxiao Wang, Binbin Lin, Xiaofei He, Xu Shen, and Jieping Ye. Enhancing multiple dimensions of trustworthiness in llms via sparse activation control. Advances in Neural Information Processing Systems, 37:15730–15764, 2024.

---

> ### Author Response · Authors · 2025-11-28
>
> Dear Reviewer qXRg,
>
> The deadline for the rebuttal phase is fast approaching. We would be grateful if you could provide us with your feedback on our responses. If you feel that our responses adequately address the concerns raised in your initial review, we would be extremely grateful for improving the score.
>
> Thank you very much for your time and contributions.
>
> Best regards, the Authors.

---

> ### Author Response · Authors · 2025-12-02
>
> Dear Reviewer qXRg,
>
> Thank you for your valuable feedback. We have revised the manuscript in response to your comments. We hope the updated version addresses your concerns.

---

### Meta-Review · Area_Chair_Feon · 2026-01-01

**Summary:**

No significant strength of the paper was recognized by the reviewers. I recommend rejecting this paper.

**Reviewer Concerns:**

* Although Reviewer qXRg gave a positive score, no significant technical details or evidence were presented to support accepting this paper.

* Both Reviewer qXRg and m7Sg pointed out that the judge model (Qwen3-8B) is not sufficient.

* Although Reviewer m7Sg also gave a positive score, the mentioned single strength is about the high-level direction of the work. On the other hand, the reviewer raised many detailed issues, and some of them were not perfectly resolved by the authors.

**Reviewer Scores:**

* Reviewer qXRg: I do not think the reviewer would have increased the score.

* Reviewer m7Sg: I do not think the reviewer would have increased the score.

* Reviewer Yss7: I do not think the reviewer would have increased the score.

* Reviewer cP2g: I do not think the reviewer would have increased the score.

---

### Decision · Program_Chairs · 2026-01-26

Reject